# Novel deep neural network architecture fusion to simultaneously predict short-term and long-term energy consumption

**Abrar Ahmed[1], Safdar Ali[1], Ali Raza[1], Ibrar Hussain[1,2], Ahmad Bilal[1], Norma Latif Fitriyani[3], Yeonghyeon Gu[3]\*, Muhammad Syafrudin[3]\***

**1** Department of Software Engineering, University of Lahore, Lahore, Pakistan, **2** Faculty of Engineering & Information Technology, Shinawatra University, Bangtoey Samkhok, Pathum Thani, Thailand, **3** Department of Artificial Intelligence and Data Science, Sejong University, Seoul, Republic of Korea

\* yhgu@sejong.ac.kr (YG); udin@sejong.ac.kr (MS)

**Data Availability Statement:** The dataset is available at the UCI Machine Learning Repository: https://doi.org/10.24432/C58K54.

## Abstract

Energy is integral to the socio-economic development of every country. This development leads to a rapid increase in the demand for energy consumption. However, due to the constraints and costs associated with energy generation resources, it has become crucial for both energy generation companies and consumers to predict energy consumption well in advance. Forecasting energy needs through accurate predictions enables companies and customers to make informed decisions, enhancing the efficiency of both energy generation and consumption. In this context, energy generation companies and consumers seek a model capable of forecasting energy consumption both in the short term and the long term. Traditional models for energy prediction focus on either short-term or long-term accuracy, often failing to optimize both simultaneously. Therefore, this research proposes a novel hybrid model employing Convolutional Neural Network (CNN), Long Short-Term Memory (LSTM), and Bi-directional LSTM (Bi-LSTM) to simultaneously predict both short-term and long-term residential energy consumption with enhanced accuracy measures. The proposed model is capable of capturing complex temporal and spatial features to predict short-term and long-term energy consumption. CNNs discover patterns in data, LSTM identifies long-term dependencies and sequential patterns and Bi-LSTM identifies complex temporal relations within the data. Experimental evaluations expressed that the proposed model outperformed with a minimum Mean Square Error (MSE) of 0.00035 and Mean Absolute Error (MAE) of 0.0057. Additionally, the proposed hybrid model is compared with existing state-of-the-art models, demonstrating its superior performance in both short-term and long-term energy consumption predictions.

## Introduction

The energy demand is steadily rising each day, while the resources available for energy production remain limited. This imbalance in distribution leads to a rapid depletion of global energy reserves, aggravated by ineffective patterns of energy consumption [1, 2]. This challenge

**Funding:** This work was partly supported by Institute of Information & communications Technology Planning & Evaluation (IITP) grant funded by the Korea government(MSIT) (No.RS-2021-II210755, Dark data analysis technology for data scale and accuracy improvement.

**Competing interests:** The authors have declared that no competing interests exist.

compels societies to grapple with accessing these resources under sustainable and reliable conditions. In such circumstances, nations face the dilemma of either enduring sluggish economic growth or opting for accelerated growth through imports, inevitably driving up energy prices and affecting budget equilibrium. This dynamic is a major indicator of many economic challenges, often linked to a growing current account deficit [3]. The U.S. Energy Information Administration (EIA) reports that the energy demand is expected to increase by 28% globally in 2040 [4], highlighting the urgent need for economies to establish a reliable energy forecasting system. Such a system is essential to proactively address the challenges associated with both consumers and industry, offering a strategic approach to navigate the complexities of energy distribution and consumption [5, 6].

Prediction of accurate energy demands is vital for the residential sector because it facilitates efficient planning and management of energy resources, empowering consumers to make informed decisions and optimize peak-hour energy consumption [7, 8]. Similarly, in the industrial sector, accurate prediction of energy consumption supports timely and cost-effective energy management, ensuring that businesses can align their energy usage efficiently for commercial operations [9–12]. Therefore, even a small improvement in optimal energy usage in both residential and industrial contexts can lead to significant reductions in overall expenses. Moreover, for energy-producing companies, precise prediction of consumption allows them to optimize generation in alignment with consumer needs. This optimization helps eliminate energy wastage during periods of low demand and prevents underproduction during peak demand, ultimately mitigating system overload [13–16].

In addressing the challenges linked to forecasting demand for energy, two crucial aspects come to the forefront: short-term and long-term energy consumption. Short-term concerns immediate and day-to-day fluctuations in energy usage, influenced by factors such as weather conditions, daily routines, and real-time demand variations. This prediction is important for load balancing, cost efficiency, and customer engagement [13, 17]. On the other hand, long-term is linked to broader trends and developments affecting energy consumption over extended periods. This includes factors like population growth, technological advancements, shifts in industrial practices, and evolving energy policies, all contributing to shaping the overall trajectory of energy demand over an extended timeframe (involves a week, month, or year) [18, 19]. Therefore, simultaneous prediction of both short-term and long-term energy consumption is essential for effective resource management. It aids in infrastructure planning, facilitates renewable integration, and supports demand-side management.

The Deep Learning (DL) approaches, such as CNN, LSTM, and Bi-LSTM, have demonstrated significant potential for accurate and reliable forecasting of energy consumption [1, 20]. The recent literature has also experimented with time series data to provide effective solutions for the challenging task of accurate energy prediction [3, 21, 22]. However, recent literature may suffer from multiple limitations, often focusing on improving either short-term or long-term energy predictions instead of both simultaneously. Additionally, the evaluation approaches employed in these studies may be limited [13, 17–19, 22, 23]. This presents a critical challenge for comprehensive energy forecasting strategies, as the focus on enhancing one aspect may compromise the accuracy of predictions in the other. Striking a balance between short-term responsiveness and long-term trend accuracy becomes imperative for a holistic and effective energy forecasting framework.

## Research questions

The below-listed Research Questions (RQs) are presented to address these limitations.

- **RQ1:** Can hybrid forecasting approaches be enhanced by integrating CNN, LSTM, and Bi-LSTM to simultaneously improve the short-term and long-term prediction of energy consumption?

- **RQ2:** How can limitations stemming from restricted or minuscule datasets be mitigated?

- **RQ3:** Can the evaluation methods and measures in this hybrid model be improved to ensure reliable and accurate predictions?

- **RQ4:** How can the integration of spatial features and time-series data be leveraged to comprehensively enhance the reliability and accuracy of the energy forecasting model?

Therefore, this research introduces a novel deep-learning model consisting of CNN, Bi-LSTM, and LSTM. The model excels in generating highly accurate results for both short-term and long-term energy consumption forecasting in residential environments. This proficiency is achieved by comprehensively understanding the spatial and temporal relations that influence energy consumption patterns. For instance, it is capable of comprehending complex spatial patterns, like the influence of building layout and occupancy distribution on energy consumption, utilizing the CNN layer. Following this, the LSTM is employed to recognize long-term dependencies and sequential patterns within time series data. Whereas, the Bi-LSTM excels in capturing intricate temporal relations in depth, leveraging its dual-processing capability, specifically tailored for handling temporal dynamics. The comparative analysis of this proposed approach, when evaluated against recent literature in the same domain, substantiates its superior performance over existing methods in accurately and simultaneously forecasting both short-term and long-term energy consumption. The effectiveness and precision of the integrated CNN, Bi-LSTM, and LSTM architecture lie in their capacity to capture both temporal and spatial relations, enhancing prediction accuracy. With this framework, we assert that utility businesses, decision-makers, and customers can derive substantial benefits, including optimized consumption, enhanced sustainability predictions, and a meaningful contribution to a more sustainable future.

## Study contributions

The following are the key contributions of this proposed research work.

- We introduce a novel deep-learning model consisting of CNN, Bi-LSTM, and LSTM. The proposed model addresses the limitation of existing literature by excelling in accuracy for both short-term and long-term energy prediction through the comprehensive capture of both temporal and spatial relations in a residential context.

- This research identifies the critical need for a hybrid forecasting framework to simultaneously predict short-term and long-term energy consumption with high accuracy.

- For the evolution of the results of this proposed framework for both long-term and short-term energy consumption forecasting, various metrics such as MAE, MSE, RMSE, and MAPE are employed. These metrics highlight the exceptional performance of the model, establishing its superiority over counterparts within the same domain.

The rest of the manuscript is divided as follows: Section 2 presents related work and research questions. Section 3 explains the components of the proposed methodology, while the results of the proposed approach are presented in Section 4. Limitations of this proposed research work are described in Section 5 and Section 6 concludes this research work.

## Literature analysis

The traditional literature in this domain presents several techniques for forecasting energy consumption, categorized into three distinct temporal intervals: short-term, long-term, and hybrid short-long-term methods. In this section, these methods are discussed in the same respective order.

### Short-term energy consumption prediction

While the majority of research studies in the literature concentrate on analyzing the behavior and occupancy patterns derived from sensor data to predict household energy consumption [24], the influence of individual attributes on the amount of power consumption may have been overlooked. Additionally, the author did not conduct a comparative analysis with traditional models, relying solely on a single accuracy measure. The prediction of building energy consumption using the firefly algorithm has been explored [25]. However, this model may fall short in achieving high accuracy, as it yielded a MAPE of 7.04%. Another notable drawback of this firefly-inspired approach is its reliance on a very small dataset. In [5], a hybrid model named EEMDRF, combining random forest (RF) and empirical mode decomposition (EEMD), is introduced for the prediction of daily energy consumption. While the EEMDRF model demonstrates strong performance in forecasting short-term energy consumption, its results for long-term energy consumption prediction may have been less satisfactory.

Furthermore, another study [26] presented a novel vector field-based regression technique. The results underscore the Support Vector Regression (SVR) model's notable precision, robustness and generalization capabilities for forecasting energy consumption. However, challenges emerge when applying this technique, particularly in the context of a smart grid system, and it is suggested that improving effective time planning, management and energy conservation practices could enhance the overall efficacy of this model. As argued by [27], the accurate prediction of short-term electrical energy consumption offers numerous advantages, including efficient resource utilization and the reduction of costs and $CO_2$ emissions.

Another study [5] introduced a hybrid stacking method to predict energy consumption. Despite successfully forecasting short-term energy consumption, it may have lacked in enhancing accuracy for long-term predictions. A hybrid model [28] introduced an innovative energy forecasting method that combined Artificial Neural Network (ANN) and SVR. The approach demonstrated superior accuracy in short-term energy consumption forecasting when compared to other similar models. Albeit, it demonstrated limited performance when predicting long-term energy consumption.

Moreover, the studies [7, 13, 29] presented a comprehensive exploration of hybrid models inspired by deep learning for the prediction of energy consumption. These models, which include GRU, CNN-GRU and DRNN-GRU, have been experimented with different sampling rates and datasets. The GRU model [13] excels in forecasting electrical energy consumption, particularly with one-minute sampling rate datasets. However, it falls short in long-term energy consumption prediction. The CNN-GRU model [29] integrates CNN and GRU in a hybrid sequential learning framework, demonstrating superior performance in computational complexity, efficiency and prediction accuracy for short-term energy consumption. However, its effectiveness for medium and long-term energy consumption remains to be tested. The DRNN-GRU model [7] leverages deep learning techniques to forecast energy demand in residential buildings using one-hour interval datasets. It effectively fills missing values in the dataset and forecasts both aggregated and disaggregated load demand in household buildings, producing high accuracy in experiments compared to other models. The ELM model [30] utilized the thermal response time of a building as a predictive model for the future. Research [8]

employed a method grounded in deep learning neural networks and ResBlock, focusing on learning correlations among various electricity load behaviors. It is important to note that the ResBlock model, while effective for short-term predictions, lacks evaluation for forecasting long-term predictions. Additionally, it utilized a small dataset for conducting experiments.

It is evident from the literature that several others, as also observed in [4, 11, 31–33], share a common deficiency in addressing long-term energy prediction. These models exhibit a notable proficiency in short-term forecasting, while they may be lacking in concurrently forecasting long-term energy consumption.

## Long-term energy consumption prediction

A research work [34] introduced an algorithm based on the hidden Markov model analyzing the residential data from a four-story building in South Korea to predict energy consumption. This model's effectiveness was then compared with Support Vector Machines, Artificial Neural Networks and Classification and Regression Trees. The results underscored the superior predictive accuracy of the proposed model, surpassing the other models by 2.96%. In [35], the authors crafted a CatBoost-based model to accurately predict long-term energy consumption, through collecting heterogeneous data from the residential building. The model achieved an impressive 99.32% accuracy and proved instrumental in identifying outliers, supporting early hazard detection and enabling the formulation of strategies to enhance energy efficiency. The research [36] introduced a hybrid LSSVIM-AIM model, integrating a least square support vector machine with an autoregressive integrated moving average. The experimental results demonstrated that the LSSVIM-AIM model produced more realistic and reliable predictions of long-term energy consumption. This complexity arises from dividing the system into subparts. A machine learning framework [37] is presented for predicting charging demand. However, this study requires extension through the analysis of charging behavior on both public and residential charging stations.

Additionally, in [38], the research introduced an SVM model for forecasting monthly electrical consumption, achieving an impressive accuracy of up to 96.8% and an F-measure of 97.4%. This performance notably surpassed other conventional approaches. Enhancing the sampling algorithm for imbalance classification can further improve the accuracy and efficiency of the SVM model. In another study [14], researchers conducted an investigation employing the Intelligent Trading Analytic Model (ITAM) to deliver real-time support and day-ahead control through a blockchain-based platform. The findings indicated that the ITAM model effectively facilitated energy crowdsourcing among prosumers and consumers, contributing to service quality. However, it's worth noting that the ITAM model's applicability is confined to long-term forecasting, and no evaluation has been conducted for short-term forecasting. In [21], the authors introduced an SVM-based PSO-SVR algorithm that incorporated data mining techniques and time series for predicting complex patterns and nonlinear data. The findings demonstrated that both the PSO-SVR and hybrid approaches yielded considerably more accurate results when compared to benchmark models like ARIMA and ANN methods. Further enhancements to PSO-SVR were observed by applying univariate methods to a multivariate dataset. In a research study [18], a D-FED deep learning model was introduced for forecasting electricity demands based on historical data. The results revealed the superior performance of the D-FED model when compared to ANN, RNN, and SVM. It's worth noting that this D-FED model underwent experimentation exclusively for long-term predictions and was not assessed for short-term energy consumption forecasting.

Moreover, in a research study [18], researchers presented a D-FED deep learning model for forecasting electricity demands using historical data. The results revealed the D-FED model's

superior performance compared to ANN, RNN and SVM. Notably, the D-FED model underwent experimentation exclusively for long-term predictions and was not assessed for short-term energy consumption forecasting. The research introduced a hybrid model (FS-FCRBM-GWDO) for energy consumption, incorporating a rectified linear unit (ReLU) activation function and a multivariate autoregressive technique for network training [19]. The FS-FCRBM-GWDO model underwent evaluation exclusively for long-term energy consumption and there is a need to extend the evaluation to include short-term energy consumption as well. In a research study [2], researchers presented a kCNN-LSTM deep learning-based framework to analyze energy consumption data and accurately predict energy consumption. Experimental results demonstrated that the kCNN-LSTM model outperformed other energy consumption forecast models. Notably, the kCNN-LSTM model underwent exclusive experimentation for long-term predictions, emphasizing the need to expand the evaluation to include forecasting short-term energy consumption.

In addition to that, research presented in [39] developed a deep learning model based on Empirical Mode Decomposition (EMD). This model seamlessly combines LSTM and EMD methods for accurate long-term energy demand forecasting. By integrating the analysis of seasonal data and implementing dynamic learning through adjustments in input and output windows during both training and testing, it excels in providing reliable predictions for extended periods. However, its efficacy diminishes when applied to shorter durations, revealing a limitation in accurately forecasting energy demands on a more immediate timescale. The authors in [16] proposed an attention-based encoder-decoder network for short-term energy prediction using the Bayesian optimization algorithm. However, it is worth noting that the research may targeted towards long-term forecasting, as the model has not been assessed for its effectiveness in short-term energy prediction.

Likewise, several other studies [12, 40–44] have been proposed, focusing primarily on enhancing the prediction of long-term energy consumption and demands. However, these models may exhibit limitations when applied to short-term energy consumption forecasting.

## Both short-term and long-term energy consumption prediction

Previously discussed literature was primarily directed towards either short-term or long-term energy prediction. However, in the existing literature, hybrid approaches can also be identified, aiming to simultaneously address both short-term and long-term energy forecasting. For instance, as suggested by [45], energy load forecasting for the smart grid is essential to meet energy demands. The author presented deep learning models to fulfill this requirement; however, it is worth noting that the evaluation may have neglected other important accuracy measures such as RMSE, MSE, MAE, etc. In [3], the authors presented a hybrid model inspired by CNN and LSTM. They employed CNN for extracting complex features from multiple variables and LSTM for modeling irregular time series data. According to their claim, the CNN-LSTM model achieved the best results compared to the findings reported in the literature within a similar domain. In another research [25], authors developed a framework named EECP-CBL for the prediction of electrical energy consumption using Bi-LSTM and CNN, which claimed to outperform other similar approaches. Nevertheless, the performance of the EECP-CBL model needs improvement through the incorporation of multiple methods, such as evolutionary algorithms and optimized versions of EECP-CBL.

Furthermore, in another research [46], a neuro-evaluation-based approach incorporating a genetic algorithm was proposed to discover the optimal set of hyperparameters for configuring deep neural networks. This neuro-evaluation-based approach achieved relatively better performances, as claimed by the authors. Albeit, the evaluation of the framework may be limited, as

only a few evaluation measures were utilized. In [23], a framework for short-term and long-term energy prediction using a deep recurrent neural network (DRNN) to capture non-linearity has been presented. The claimed results demonstrated that the DRNN provides 20–45% improvement in energy prediction compared to other conventional models for both short-term and long-term load prediction. However, there remains a considerable gap to enhance the accuracy of the prediction. An interesting approach in [22] introduced an online adaptive RNN capable of continuously learning real-time data and updating weights based on the current input state. The claimed results demonstrate that the RNN method achieved higher accuracy compared to traditional offline short-term and long-term memory prediction models in the existing literature. In [24], residential energy consumption has been predicted using Artificial Neural Networks, and the claimed results achieve exceptional accuracy and the lowest mean square error (MSE) compared to conventional machine learning models.

The above review of related work identifies several limitations. Primarily, the hybrid approaches, which simultaneously forecast both short and long-term energy consumptions, often excel at only one aspect: either improving short-term or long-term predictions, but not both concurrently. While a few approaches claimed to have enhanced both aspects, limitations may arise due to restricted/minuscule datasets or evaluation methods/measures. Additionally, the integration of spatial features and time-series data into models may not be comprehensive, thereby limiting reliability and effectiveness.

## Proposed methodology

In this study, we adopted a stepwise methodology to predict power consumption more efficiently, as shown in Fig 1. Initially, a comprehensive power consumption dataset is acquired, serving as the foundational input for our analysis. Following data acquisition, normalization is

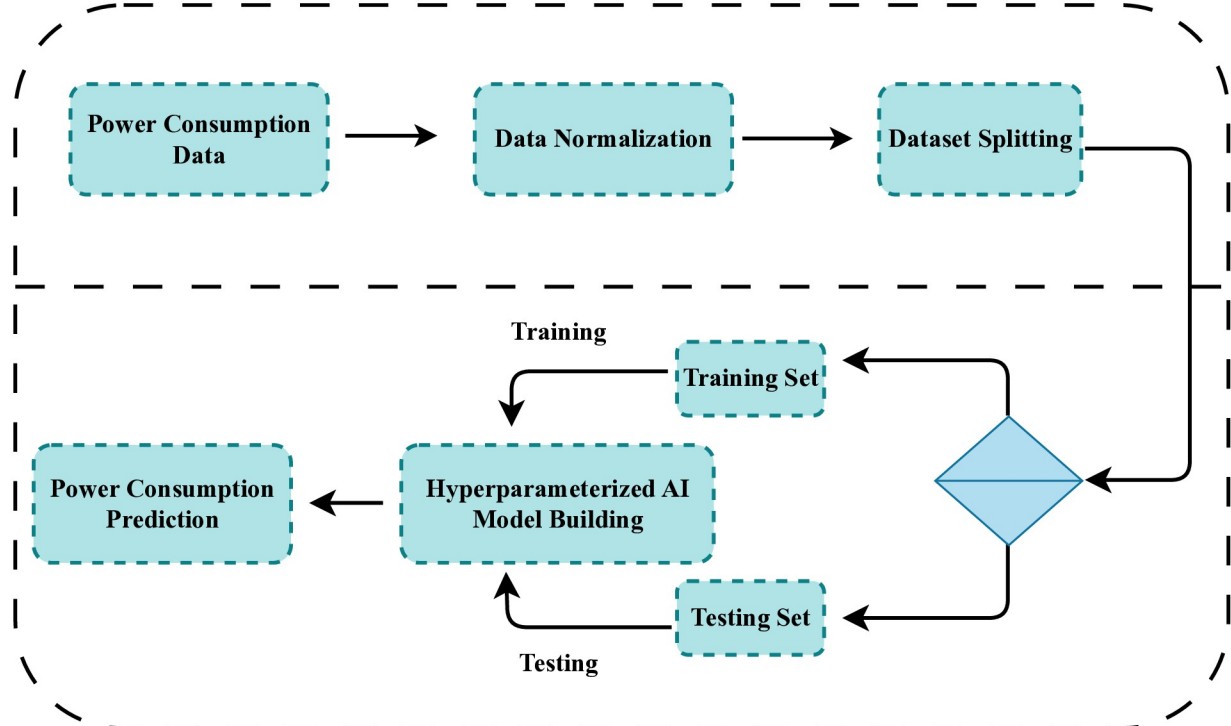

**Fig 1. The workflow of the proposed methodology.**

performed to ensure the dataset is scaled appropriately, thereby eliminating any biases due to varying data ranges. Post-normalization, the dataset is partitioned into two distinct subsets: a training set and a test set. The training set was utilized to train an advanced deep-learning model, leveraging its capacity to learn intricate patterns and relationships within the data. Subsequently, the trained model is evaluated using the test set to assess its performance and generalizability. The proposed model demonstrated improved efficiency in predicting power consumption, highlighting its potential for practical applications in energy management and optimization.

## Dataset acquisition

The proposed research work utilizes the Individual Household Electric Power Consumption (IHEPC) dataset, and it has also been employed by recent literature in the same domain [4, 17, 20, 47]. This dataset spans a four-year period, from 2006 to 2010, and is available at the UCI Machine Learning Repository [3]. With a substantial size of 2,075,259 records and encompassing 12 attributes, the IHEPC dataset serves as a comprehensive resource for exploring energy consumption patterns. The following are the 12 attributes of this dataset:

1. Minute: A value ranging from 1 to 60.

2. Hour: A value ranging from 0 to 23.

3. Day: A value ranging from 1 to 31.

4. Month: A value ranging from 1 to 12.

5. Year: A value ranging from 2006 to 2010.

6. Global Active Power: The household global minute-averaged active power (in kilowatts).

7. Global Reactive Power: The household global minute-averaged reactive power (in kilowatts).

8. Voltage: The minute-averaged voltage (in volts).

9. Global Intensity: The household global minute-averaged current intensity (in amperes).

10. Sub Metering 1: This variable corresponds to the kitchen, mainly containing a dishwasher, an oven and a microwave, with hot plates powered by gas (in watt-hours of active energy).

11. Sub Metering 2: This variable corresponds to the laundry room, containing a washing machine, a tumble-dryer, a refrigerator and a light (in watt-hours of active energy).

12. Sub Metering 3: This variable corresponds to an electric water heater and an air conditioner (in watt-hours of active energy).

## Data normalization

Data normalization is a crucial preprocessing step in the analysis of power consumption datasets, ensuring that the data is transformed into a common scale without distorting differences in the ranges of values. This process is particularly significant in power consumption data, which often encompasses diverse scales due to varying measurement units and operational contexts. This is especially beneficial for machine learning algorithms sensitive to the scale of input features, such as gradient descent-based optimization. Normalization not only enhances the performance of these algorithms but also improves the interpretability of statistical

analyses and visualizations, leading to more robust and generalizable insights into power consumption patterns.

## Data visualization

In the following Fig 2, a correlation matrix analysis is given. In the following table, positive 1 indicates there is a positive correlation, -1 indicates there is a negative correlation and 0 indicates there is no correlation, the first matrix shows the correlation of minute analysis, whereas the second matrix shows the correlation of hourly data analysis, the third matrix shows correlation matrix of daily data and last matrix shows correlation matrix of weekly data.

## Novel proposed neural network architecture fusion

The proposed architecture effectively integrates CNN, LSTM, and Bi-LSTM layers, with each of these layers playing a critical role in enhancing the prediction accuracy of energy consumption for both short-term and long-term periods. All layers perform in a manner that most

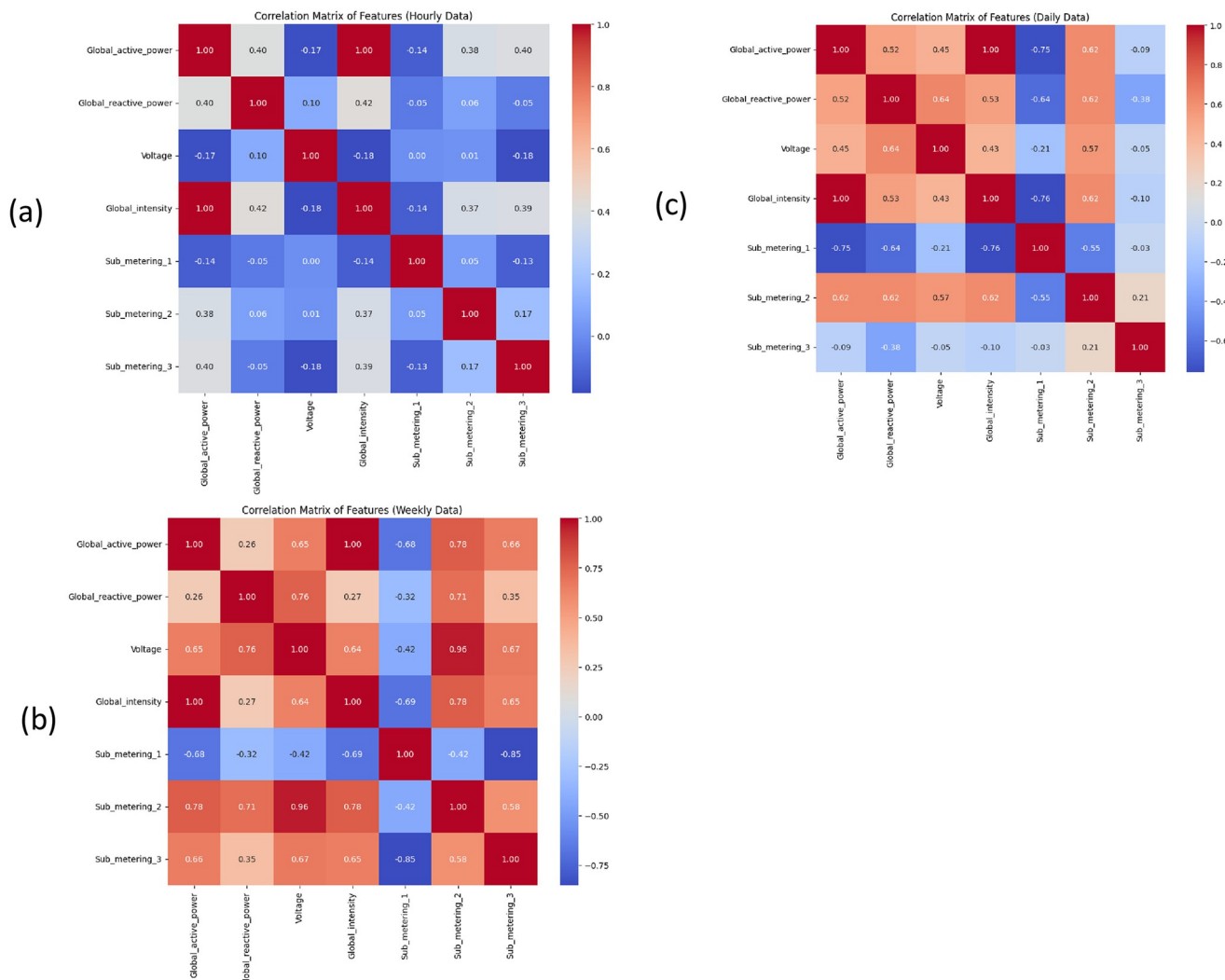

**Fig 2. A correlation matrix analysis: (a) hourly data, (b) daily data, (c) weekly data.**

**CNN+BiLSTM+LSTM**

**Fig 3. The novel proposed architecture.**

effectively addresses various components of the data, utilizing each layer's specific capabilities to produce accurate and reliable predictions of energy consumption. The CNN enables the proposed model to take the impact of physical layout (spatial information) and equipment arrangements into consideration. The Bi-LSTM component further enhances the model's capabilities by concurrently taking into account past and future contexts. The LSTM component is excellent at capturing short-term variations and temporal dependencies. The components of this proposed framework are described below and also depicted in Fig 3.

The proposed model processes sequential data using multiple unique layers, each dedicated to a specific task for extracting and learning relevant features from input sequences. The initial Conv1D layer operates on the input data. In this layer, 64 learnable filters are applied to the input sequence, screening the feature data to extract spatial features essential for predicting energy consumption. The output shape is (None, 22, 64), where "None" refers to the batch size, the window size is 22, and 64 represents the features extracted by the convolutional layer. This layer applies the Rectified Linear Unit (ReLU) activation function. ReLU introduces non-linearity by assigning zero output to negative inputs and retaining positive inputs as they are. This not only facilitates efficient learning but also helps alleviate vanishing gradient issues.

Subsequently, Layer-2 employs a MaxPooling1D to reduce the dimensionality of the input data. By using a pooling size of 2x2, the data is halved, resulting in a final shape of (None, 11, 64). This step is crucial for capturing the most important spatial features and reducing computational complexity.

Following the dimensionality reduction, Layer 3 introduces a dropout to prevent overfitting and enhance generalization. This technique randomly sets a fraction of the input units to zero during training. The output of this layer remains the same as the previous layer (None, 11, 64).

After the dropout layer, Layer 4 employs a Bi-LSTM to process the data, capturing temporal dependencies from both past and future time steps. With 256 LSTM units, the Bi-LSTM layer provides an output shape of (None, 11, 256). The output of this layer is then passed through a traditional LSTM in Layer-5 to further refine the representations and learn from the temporal

patterns captured by the Bi-LSTM. This yields hidden states with 128 dimensions for each sequence, resulting in an output shape of (None, 128).

Finally, in Layer-6, a dense layer is applied to generate the final predictions, with an output shape of (None, 7). This layer is responsible for mapping the output to the desired prediction space.

The model is compiled (trained) using the MSE loss function to measure performance, with the "Adam" optimization algorithm updating weights to minimize the loss. The learnable parameters are fine-tuned at each layer to minimize prediction errors through the automated process of Adam adjusting weights and biases.

The proposed model is represented by Eq 1, as given below.

$$y(t) = \text{Dense}(\text{LSTM}(\text{BiLSTM}(\text{CNN}(X)))) \tag{1}$$

In Eq 1, $y(t)$ represents the final prediction at a time 't'. The 'X' denotes input data organized into batches with a size of 32 and a time window of 24 steps. The input_dim, specified by input_shape = (window_size, X_train.shape [2]), indicates that 'X' has 2 features at each time step 't'.

$$x = \text{CNN}(W_{\text{input}} \cdot X + b_{\text{input}}) \tag{2}$$

$$h_{\text{BiLSTM}} = \text{BiLSTM}(W_{\text{CNN}} \cdot x + b_{\text{CNN}}) \tag{3}$$

$$h_{\text{LSTM}} = \text{LSTM}(W_{\text{BiLSTM}} \cdot h_{\text{BiLSTM}} + b_{\text{BiLSTM}}) \tag{4}$$

$$y(t) = \text{Dense}(W_{\text{LSTM}} \cdot h_{\text{LSTM}} + b_{\text{LSTM}}) \tag{5}$$

In Eq 2, 'X' denotes the input data of shape (batch-size, window_size, input_dim). The CNN layer is applied to 'X' (represented by small 'x' in Eq 2) using weights ($W_{\text{input}}$) and biases ($b_{\text{input}}$). The output, denoted as $h_{\text{BiLSTM}}$ in Eq 3, is then processed by the Bi-LSTM layer. The Bi-LSTM layer has weights ($W_{\text{CNN}}$) and biases ($b_{\text{CNN}}$). The output, $h_{\text{BiLSTM}}$, is further fed into the LSTM layer ($h_{\text{LSTM}}$ in Eq 4), which comprises weights ($W_{\text{BiLSTM}}$) and biases ($b_{\text{BiLSTM}}$). The final prediction at time step 't' is obtained by extracting the output of the LSTM layer, denoted as $h_{\text{LSTM}}$. This output is generated by a dense layer represented by $y(t)$ with weights ($W_{\text{LSTM}}$) and biases ($b_{\text{LSTM}}$) in Eq 5. This mathematical representation illustrates the sequential flow of operations in the model. Each layer's output, calculated with respective weights and biases, serves as the input to the next layer, culminating in the generation of the final prediction $y(t)$.

## Applied deep learning model layer configuration

Initially, the proposed framework incorporates CNNs into the model, facilitating the extraction of crucial spatial representations such as building layouts, occupancy patterns, and the spatial placement of appliances within residential areas. This integration of geographical details empowers the model with a deeper understanding of the data, consequently enhancing its ability to predict energy consumption.

Subsequently, a Bi-directional LSTM (Bi-LSTM) layer is integrated to further augment the model's predictive capability. By concurrently considering both preceding information and future context, Bi-LSTM enhances the advantages offered by LSTM. The model adeptly captures both previous and forthcoming temporal contexts by processing the input sequence bidirectionally through two LSTM sub-layers. This dual processing enables the model a capture a deep understanding of the intricate temporal connections inherent in energy consumption. Consequently, the model attains enhanced prediction accuracy, particularly in scenarios

characterized by long-term dependencies or where precise predictions are significantly influenced by future contextual information.

Following that, this framework architecture employs Recurrent Neural Networks (RNNs) [48–51] with Long Short-Term Memory (LSTM) to identify long-term dependencies and sequential patterns within time series data. They address the vanishing gradient issue, allowing for the retention and utilization of relevant historical data over extended periods, overcoming limitations seen in traditional RNNs. The proposed framework effectively captures dependencies and patterns inherent in energy consumption by integrating LSTM layers into its architecture. This integration enables the model to consider both the most recent and historical energy usage when formulating predictions. The approach enhances accuracy by accounting for short-term fluctuations and trends, showcasing the model's ability to grasp the intricacies of energy consumption dynamics.

Overall, this model processes sequential data by employing a combination of Conv1D, MaxPooling1D, Dropout, Bi-LSTM, LSTM and Dense layers, and the total number of trainable parameters is 397,063. The proposed model is trained to minimize prediction errors using a suitable loss function and optimization algorithms, enabling it to make accurate predictions for the given time-dependent task. The configuration details of the proposed model are outlined in Table 1.

## Results and discussions

In this paper, the proposed model is analyzed to compare its prediction accuracy using evaluation measures such as MSE, RMSE, MAE, and MAPE. The dataset is divided into a 60–40 split, where 60% is employed as training data and the remaining 40% is reserved for testing purposes.

### Experimental setup

The research experiments are conducted using Google Colab, and the computations are performed through freely available GPUs on the Google Colab platform (S1 Code). For training and testing sets, we use a 70:30 split ratio. In detail, 70% of the dataset is set out for training the model, whereas the rest 30% is utilized for testing and validation. In addition, 5-fold cross-validation is practiced to ensure the robustness of the proposed model. The model is implemented using Keras with TensorFlow as the backend, utilizing Python for data processing and model training. The hyperparameters used for the proposed models are expressed in Table 2.

The first metric utilized is Mean Squared Error (MSE), as represented in Eq 6. MSE calculates the square root of the average of the errors, specifically the average squared difference

**Table 1. Layer configuration details of the proposed model.**

| Layer (type). | Output Shape | Param # |
|---|---|---|
| 1—conv1d (Conv1D). | (None, 22, 64) | 1408 |
| 2—max_pooling1d (MaxPooling1D). | (None, 11, 64) | 0 |
| 3—dropout (Dropout). | (None, 11, 64) | 0 |
| 4—bidirectional (Bidirectiona1). | (None, 11, 256) | 197632 |
| 5—lstm_1 (LSTM). | (None, 128) | 197120 |
| 6—dense (Dense). | (None, 7) | 903 |
| Total params: | 397,063 | |
| Trainable params | 397,063 | |

**Table 2. The hyperparameters used for the models.**

| Models | Hyperparameters Description |
|---|---|
| CNN | Number of filters: 64, Kernel size: 3, Activation function: ReLU, Pooling size: 2, Dropout rate: 0.2 |
| LSTM | Number of units: 128, Return sequences: True for the first LSTM layer, False for the second |
| Bi-LSTM | Number of units: 128, Optimizer: The Adam optimizer was chosen due to its adaptive learning rate feature for faster convergence. Loss Function: The MSE was selected as the loss function to find the best JSON model. Batch Size: 32, Epochs: The model was trained for 100 epochs, with early stopping applied based on the validation loss (patience of 10 epochs). |

between the expected and actual values.

$$\text{MSE} = \frac{1}{n}\sum_{i=1}^{n}(D_i - \hat{D}_i)^2 \tag{6}$$

The standard deviation of prediction errors is referred to as Root Mean Squared Error (RMSE). It considers the measurement of how distant the data points are from the regression line, making it a metric of how evenly dispersed these residuals are. This metric, determined as follows in Eq 7, is commonly employed in climatology, prediction, and regression analysis to validate experimental models.

$$\text{RMSE} = \sqrt{\frac{1}{n}\sum_{i=1}^{n}(D_i - \hat{D}_i)^2} \tag{7}$$

Additionally, Mean Absolute Error (MAE) is employed for evaluation. MAE considers the average size of prediction errors while disregarding their directions. The value is defined as the average of the absolute differences between predicted and actual values for each instance in the testing set. MAE assigns equal weight to all individual variances. Eq 8 provides the detailed expression for calculating MAE.

$$\text{MAE} = \frac{1}{n}\sum_{i=1}^{n}|y_i - \hat{y}_i| \tag{8}$$

Finally, the Mean Absolute Percentage Error (MAPE) measures how well a predicting technique performs. Eq 9 provides the accuracy expressed as a percentage for this metric.

$$\text{MAPE} = \frac{1}{n}\sum_{i=1}^{n}\left|\frac{D_i - \hat{D}_i}{D_i}\right| \times 100 \tag{9}$$

The proposed model is compared with existing state-of-the-art models in the literature, namely LSTM [3], CNN-LSTM [3], and CNN-Bi-LSTM [25], for minutely, hourly, daily, and weekly data. The data used in these instances of the literature is also derived from the IHEPC dataset, which is employed in this proposed model.

## Performance results of applied methods

**Results with minute data.** Table 3 presents the results of forecasting energy consumption for minute-level data. The proposed model exhibits superior performance with minimum error rates, achieving values of 0.0001, 0.011, 0.002, and 11.39 for MSE, RMSE, MAE, and MAPE, respectively. In comparison, for the same evaluation metrics, the existing state-of-the-art techniques, including EECP-CBL [6] records values of 0.051, 0.225, 0.098, and 11.66, while

**Table 3. Performance of experimental methods for minute data.**

| #No | Model | MSE | RMSE | MAE | MAPE |
|---|---|---|---|---|---|
| 1 | Linear Regression [20] | 0.405 | 0.636 | 0.418 | 74.52 |
| 2 | LSTM [20] | 0.748 | 0.865 | 0.628 | 51.45 |
| 3 | CNN-LSTM [4] | 0.374 | 0.611 | 0.349 | 34.84 |
| 4 | EECP-CBL [20] | 0.051 | 0.225 | 0.098 | 11.66 |
| 5 | Decision tree [47] | 0.59 | 0.77 | 0.54 | - |
| 6 | SVR [47] | 0.59 | 0.77 | 0.49 | - |
| 7 | CNN [47] | 0.37 | 0.67 | 0.47 | - |
| 8 | CNN-GRU [47] | 0.22 | 0.47 | 0.33 | - |
| 9 | **Proposed Model** | 0.0001 | 0.011 | 0.002 | 11.39 |

CNN-LSTM [1] yields figures of 0.374, 0.611, 0.349 and 34.84. On the other hand, the LSTM achieves 0.748, 0.865, 0.628 and 51.45. The Linear Regression output values of 0.405, 0.636, 0.418, and 74.52. The disparities between the performance of our proposed model and other approaches are notably significant, as it outperforms all other approaches.

**Results with hourly data.** Table 4 details the results of hourly-level data, where this proposed model exhibits superior performance, achieving minimum error rates with values of 0.0009 for MSE, 0.031 for RMSE, 0.019 for MAE, and 50.29 for MAPE. Other models in comparison include EECP-CBL [6], which records values of 0.298, 0.546, 0.392, and 50.09. The CNN-LSTM [1] achieves 0.355, 0.596, 0.332 and 32.83. On the other hand, the LSTM attains values at 0.515, 0.717, 0.526, and 44.37, while the Linear Regression obtains values at 0.425, 0.652, 0.502, and 83.74, respectively for the same metrics. The disparities between the performance of our proposed model and other approaches are notably significant, as it outperforms all other approaches.

**Results with daily data.** Table 5 outlines the results of daily-level data, where this proposed model demonstrates superior performance with minimum error rates, achieving values of 0.0002 for MSE, 0.016 for RMSE, 0.009 for MAE and 18.57 for MAPE, respectively. Other models in comparison include EECP-CBL [6], recording values of 0.065, 0.255, 0.191 and 19.15, while the CNN-LSTM [1] achieves 0.104, 0.322, 0.257 and 31.83. The LSTM attains values at 0.241, 0.491, 0.413 and 38.72, and Linear Regression yields values of 0.253, 0.503 0.392, and 52.69, respectively for same evaluation metrics. The disparities between the performance of our proposed model and other approaches are notably significant, as it outperforms all other approaches.

**Table 4. Performance of experimental methods for hourly data.**

| #No | Model | MSE | RMSE | MAE | MAPE |
|---|---|---|---|---|---|
| 1 | Linear Regression [20] | 0.425 | 0.652 | 0.502 | 83.74 |
| 2 | LSTM [20] | 0.515 | 0.717 | 0.526 | 44.37 |
| 3 | CNN-LSTM [4] | 0.355 | 0.596 | 0.332 | 32.83 |
| 4 | EECP-CBL [20] | 0.298 | 0.546 | 0.392 | 50.09 |
| 5 | CNN-2D-RP [17] | - | 0.79 | 0.59 | - |
| 6 | CNN-LSTM-AE [52] | 0.19 | 0.47 | 0.31 | - |
| 7 | CNN-MLBDLSTM [31] | 0.31 | 0.56 | 0.34 | - |
| 8 | SE-AE [6] | 0.38 | - | 0.39 | - |
| 9 | FCRBM [53] | - | 0.66 | - | - |
| 10 | **Proposed Model** | 0.0009 | 0.031 | 0.019 | 50.29 |

**Table 5. Performance of experimental methods for daily data.**

| #No | Model | MSE | RMSE | MAE | MAPE |
|---|---|---|---|---|---|
| 1 | Linear Regression [20] | 0.253 | 0.503 | 0.392 | 52.69 |
| 2 | LSTM [20] | 0.241 | 0.491 | 0.413 | 38.72 |
| 3 | CNN-LSTM [4] | 0.104 | 0.322 | 0.257 | 31.83 |
| 4 | EECP-CBL [20] | 0.065 | 0.255 | 0.191 | 19.15 |
| 5 | **Proposed Model** | 0.0002 | 0.016 | 0.0009 | 18.57 |

**Results with weekly data.** Table 6 presents the results of weekly-level data, where this proposed model demonstrates superior performance with minimum error rates, achieving values of 0.0002 for MSE, 0.015 for RMSE, 0.009 for MAE and 16.20 for MAPE, respectively. Other models in the comparison include EECP-CBL [6], recording values of 0.049, 0.220, 0.177 and 21.28. The CNN-LSTM [1] achieves 0.095, 0.309, 0.238, and 31.84, while the LSTM attains values at 0.105, 0.324, 0.244 and 35.78, and Linear Regression yields values of 0.148, 0.385, 0.320 and 41.33, respectively for the same evaluation metrics. The disparities between the performance of our proposed model and other approaches are notably significant, as it outperforms all other approaches.

## Time series during training analysis

Fig 4 graphs show the model loss over Epochs. For minute analysis validation loss is less than 0.005, for hourly data validation loss is 0.002, for daily analysis validation loss is near about 0.0 and for weekly data validation loss is less than 0.05. These validation losses over Epochs show that the proposed model is not overfitting and achieving very close alignment to the validation data, hence it proves that our proposed model is working well.

## Feature importance rankings analysis

Fig 5 graphs show feature importance ranking in the dataset. For minute data global_active_power has a great influence on the dataset and it contributes heavily to predicting with model due to its high correlation with prediction. For other analyses such as hourly data, daily data and weekly data, sub_metering_3 has high importance due to its predictive patterns and high correlation with target.

## Comparison with decomposition algorithms

Table 7 shows the performance of the proposed model against models incorporating decomposition algorithms, which include EMD, EEMD, and CEEMDAN. The analysis infers that

**Table 6. Performance of experimental methods for weekly data.**

| #No | Model | MSE | RMSE | MAE | MAPE |
|---|---|---|---|---|---|
| 1 | Linear Regression [20] | 0.148 | 0.385 | 0.320 | 41.33 |
| 2 | LSTM [20] | 0.105 | 0.324 | 0.244 | 35.78 |
| 3 | CNN-LSTM [4] | 0.095 | 0.309 | 0.238 | 31.84 |
| 4 | EECP-CBL [20] | 0.049 | 0.220 | 0.177 | 21.28 |
| 5 | FCRBM [53] | - | 0.79 | - | - |
| 6 | **Proposed Model** | 0.0002 | 0.015 | 0.0009 | 16.20 |

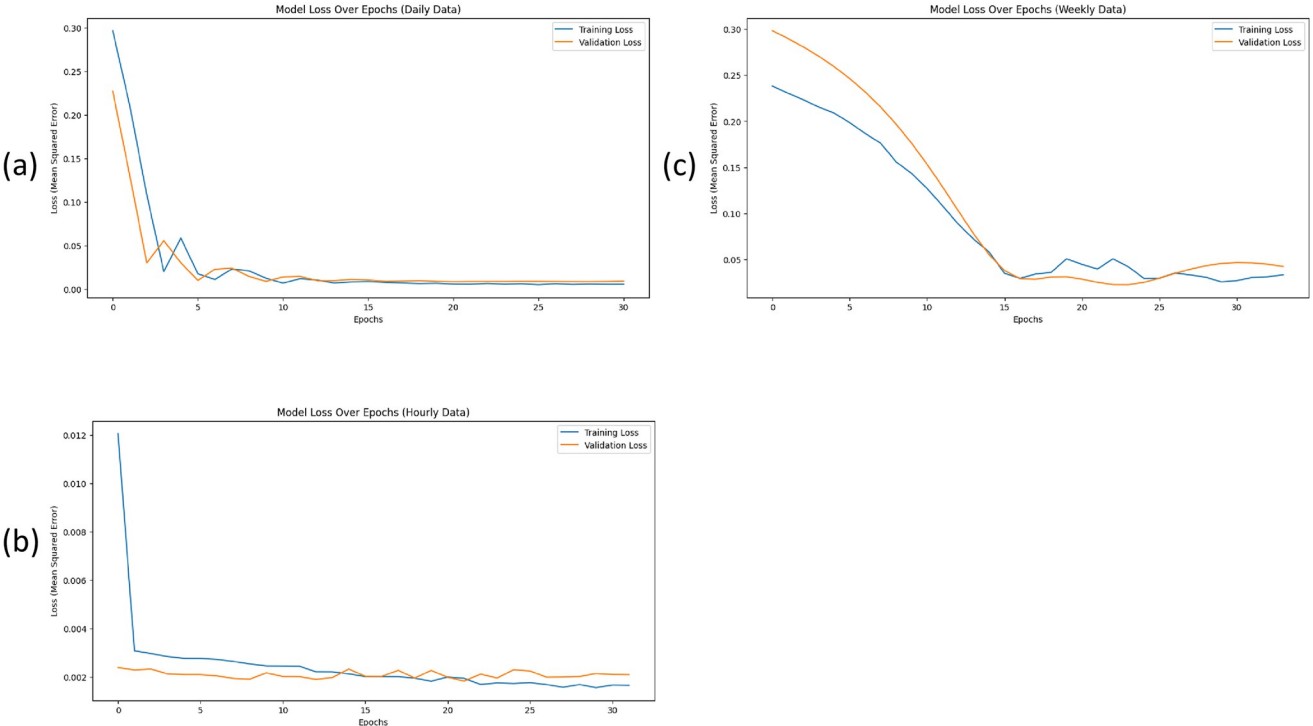

**Fig 4. The Time series during training analysis: (a) hourly data, (b) daily data, (c) weekly data.**

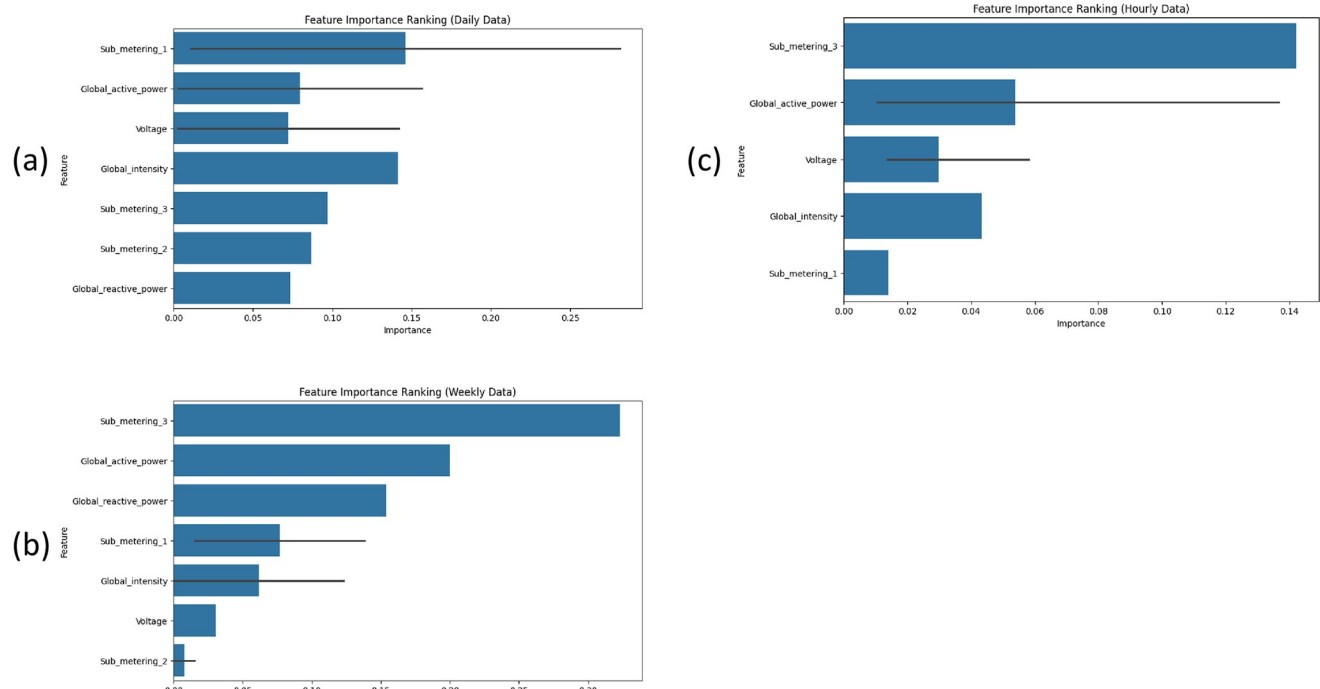

**Fig 5. The feature importance rankings analysis: (a) hourly data, (b) daily data, (c) weekly data.**

**Table 7. The comparison results with decomposition algorithms.**

| Model | Minutely | | Hourly | | Daily | | Weekly | |
|---|---|---|---|---|---|---|---|---|
| EMD | MSE | 0.9694 | MSE | 0.2946 | MSE | 9.68 | MSE | 0.0016 |
| | RMSE | 0.9846 | RMSE | 0.5428 | RMSE | 0.0098 | RMSE | 0.0408 |
| | MAE | 0.7029 | MAE | 0.4162 | MAE | 0.0098 | MAE | 0.0340 |
| | MAPE | 6670.6% | MAPE | 3955.9% | MAPE | 94.38% | MAPE | 334.2% |
| EEMD | MSE | 0.9694 | MSE | 0.2946 | MSE | 0.2017 | MSE | 0.3093 |
| | RMSE | 0.9846 | RMSE | 0.5428 | RMSE | 0.4492 | RMSE | 0.5561 |
| | MAE | 0.7029 | MAE | 0.4162 | MAE | 0.3637 | MAE | 0.4427 |
| | MAPE | 6670.6% | MAPE | 3955.9% | MAPE | 3537.3% | MAPE | 4200.2% |
| CEEMDAN | MSE | 1.0046 | MSE | 1.0081 | MSE | 1.0136 | MSE | 1.0043 |
| | RMSE | 1.0023 | RMSE | 1.0040 | RMSE | 1.0068 | RMSE | 1.0021 |
| | MAE | 0.9004 | MAE | 0.9030 | MAE | 0.9005 | MAE | 0.9003 |
| | MAPE | 9691.6% | MAPE | 9401.7% | MAPE | 8675.6% | MAPE | 8762.2% |

using each algorithm-propped model achieved optimal performance scores for predicting energy consumption with a minimum error rate.

**Performance summarization with baseline models.** Table 8 summarizes the average performance of the proposed model compared to similar techniques in the literature. The values are computed by averaging minute, hourly, daily and weekly results. Across these intervals, the proposed model consistently outperforms others based on four standard evaluation measures of MSE, RMSE, MAE and MAPE. This superiority extends to minute, hourly, daily and weekly data from the IHEPC dataset. This work can enhance intelligent power management systems, especially in predicting future electric power consumption. The average results are visualized in Fig 6.

## Computational complexity

The computational complexity of the proposed model in Table 9 demonstrates its efficiency across varying temporal granularities. Using the IHEPC dataset, the model required 486 seconds for daily data, 119 seconds for weekly data, and just 32 seconds for hourly data. This performance highlights the model's scalability and suitability for both coarse-grained and fine-grained energy consumption analysis, ensuring practical usability across diverse applications.

**Table 8. Average performance of the proposed model compared to with baseline models.**

| #No | Model | MSE | RMSE | MAE | MAPE |
|---|---|---|---|---|---|
| 1 | Linear Regression [20] | 0.30775 | 0.544 | 0.408 | 63.07 |
| 2 | LSTM-attention layer [20] | 0.40225 | 0.59925 | 0.45275 | 42.58 |
| 3 | CNN-LSTM [4] | 0.232 | 0.4595 | 0.294 | 32.835 |
| 4 | EECP-CBL [20] | 0.11575 | 0.3115 | 0.2145 | 25.545 |
| 5 | Decision tree [47] | 0.59 | 0.77 | 0.54 | - |
| 6 | SVR [47] | 0.59 | 0.77 | 0.49 | - |
| 7 | CNN-attention layer [47] | 0.37 | 0.67 | 0.47 | - |
| 8 | CNN-GRU [47] | 0.22 | 0.47 | 0.33 | - |
| 9 | Bi-LSTM [1] | 0.104 | 0.322 | 0.257 | 31.83 |
| 10 | **Proposed Model** | 0.00035 | 0.01825 | 0.0057 | 24.1125 |

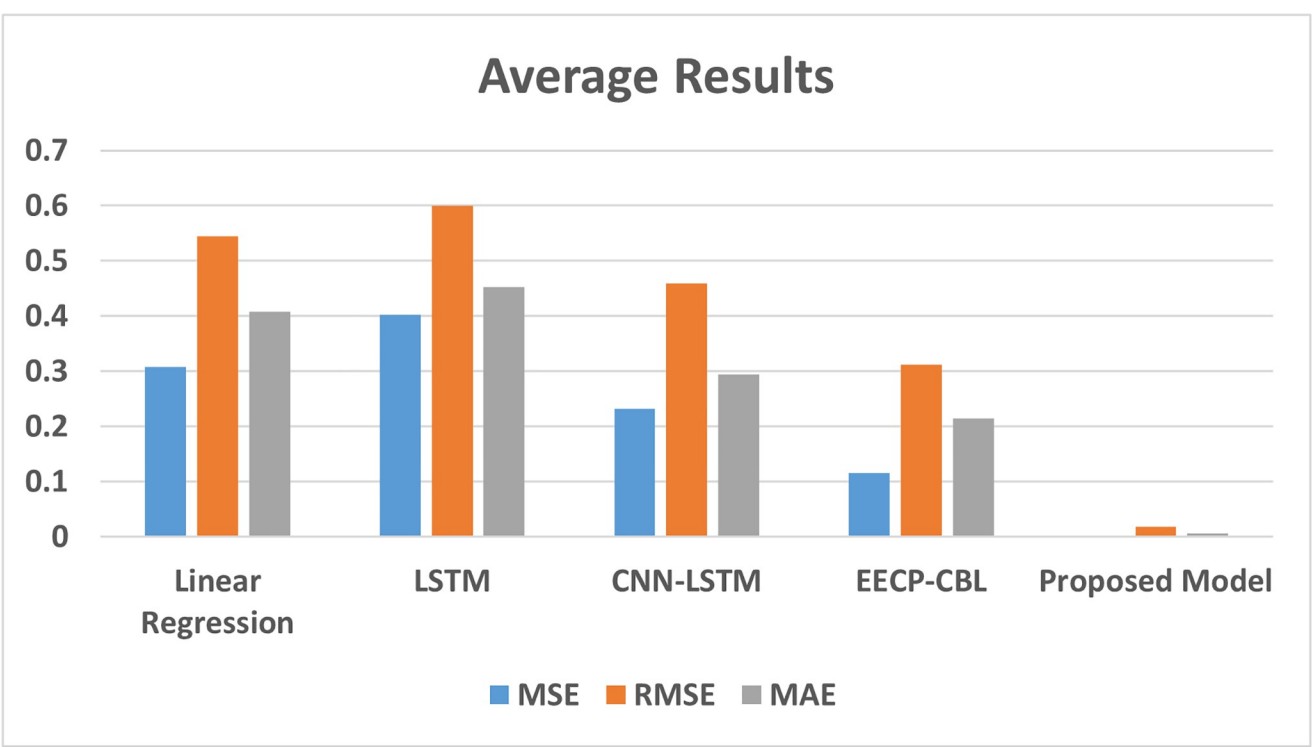

**Fig 6. Average performance of the proposed model compared to similar techniques.**

## Ablation study

In our ablation study, we conducted a comparative analysis to investigate the performance of hybrid deep learning models using a complete dataset. The results, as detailed in Table 10 for each model. The CNN-LSTM model yielded an MSE of 0.095, RMSE of 0.309, MAE of 0.238, and MAPE of 31.84%, indicating a reasonable predictive performance. However, the CNN+Bi-LSTM model, with slightly higher error metrics, did not significantly outperform the CNN-LSTM. Notably, the BiLSTM+LSTM configuration achieved superior accuracy, with minimal errors across all metrics. This significant reduction in errors suggests that BiLSTM+LSTM is highly effective for this task, capturing complex patterns more accurately compared to the other configurations.

## Conclusion and future work

This research introduces a deep learning-inspired framework designed to simultaneously forecast short-term and long-term energy consumption with high accuracy and a low error rate. The proposed model effectively captures intricate temporal and spatial features crucial for

**Table 9. The computational complexity of the proposed model.**

| Data Portions | Runtime Computations (Seconds) |
|---|---|
| For Daily Data | 486s |
| For Weekly Data | 119s |
| For Hourly Data | 32s |

**Table 10. The ablation study analysis.**

| Model | MSE | RMSE | MAE | MAPE |
|---|---|---|---|---|
| CNN-LSTM | 0.095 | 0.309 | 0.238 | 31.84 |
| CNN+Bi-LSTM | 0.104 | 0.322 | 0.257 | 31.83 |
| BiLSTM+LSTM | 0.008 | 0.090 | 0.067 | 11.51 |

**Table 11. Description of acronyms used in this manuscript.**

| Short Name | Abbreviation |
|---|---|
| EECP | Electric Energy Consumption Prediction |
| CNN | Convolutional Neural Network |
| Bi-LSTM | Bi-directional Long Short-Term Memory |
| IHEPC | Individual Household Electric Power Consumption dataset |
| LSTM | Long Short-Term Memory |
| RNN | Recurrent Neural Network |
| MSE | Mean Square Error |
| RMSE | Root Mean Square Error |
| MAE | Mean Absolute Error |
| MAPE | Mean Absolute Percentage Error |
| MLP | Multilayer Perceptron |
| ReLU | Rectified Linear Unit |
| EECP-CBL | Electric Energy Consumption Prediction model utilizing the combination of CNN and Bi-LSTM |
| RF | Random forest |
| RT | Random Tree |
| CFS | Correlation-based feature selection |
| NCA FS | Neighborhood Component Analysis Regression-based feature selection |
| MRE | Mean relative error |
| SMAPE | Symmetric mean absolute percentage error |
| EDA | Exploratory data analysis |
| IMFs | Intrinsic mode functions |
| MLR | Multiple liner regression |
| BPNN | Back-propagation neural networks |
| EDL | Extreme deep learning |
| GBR | Gradient boosting regression |
| IDT | Information diffusion technology |
| HMTD | Heuristic mega-trend diffusion |
| LSTM-AE | Auto encoder long short-term memory |
| GRU | Gated-recurrent unit |
| SD | Standard deviation |
| EMD | Empirical mode decompositions |
| FFANN | Feedforward artificial neural network |

predicting both short-term and long-term energy consumption. By employing Convolutional Neural Networks (CNNs) to identify patterns, Long Short-Term Memory (LSTM) to identify long-term dependencies and sequential patterns, and Bidirectional LSTM (Bi-LSTM) to recognize complex temporal relations within the data, our model demonstrates robust predictive capabilities. Furthermore, we benchmarked the proposed hybrid model against existing state-of-the-art models in the same domain, revealing its superior performance in predicting both

short-term and long-term energy consumption. This framework holds significant potential for optimizing energy consumption for both commercial and residential consumers, thereby contributing to sustainability efforts and fostering a more sustainable future. Looking ahead, our future work aims to enhance the model by incorporating additional contextual and geographical data, recognizing the variation in energy consumption patterns from city to city. This evolution will further refine the accuracy and applicability of our framework.

## Acronyms

The acronyms in Table 11 are provided for clarity and convenience. These acronyms are intended to facilitate clear communication in the discussion of the proposed model, related work, and evaluation metrics.

## Supporting information

**S1 Code.**
(IPYNB)

## Author Contributions

**Conceptualization:** Abrar Ahmed, Safdar Ali, Ali Raza, Ibrar Hussain, Ahmad Bilal, Norma Latif Fitriyani, Yeonghyeon Gu, Muhammad Syafrudin.

**Data curation:** Abrar Ahmed, Safdar Ali, Ali Raza, Ibrar Hussain, Ahmad Bilal.

**Formal analysis:** Abrar Ahmed, Safdar Ali, Ali Raza, Ibrar Hussain, Ahmad Bilal, Norma Latif Fitriyani, Muhammad Syafrudin.

**Funding acquisition:** Yeonghyeon Gu, Muhammad Syafrudin.

**Investigation:** Ibrar Hussain, Ahmad Bilal.

**Methodology:** Abrar Ahmed, Safdar Ali, Ali Raza, Ibrar Hussain, Ahmad Bilal, Norma Latif Fitriyani, Yeonghyeon Gu, Muhammad Syafrudin.

**Software:** Abrar Ahmed, Safdar Ali, Ali Raza, Norma Latif Fitriyani.

**Supervision:** Yeonghyeon Gu, Muhammad Syafrudin.

**Visualization:** Abrar Ahmed, Safdar Ali, Ali Raza, Norma Latif Fitriyani, Muhammad Syafrudin.

**Writing – original draft:** Abrar Ahmed, Safdar Ali, Ali Raza, Ibrar Hussain, Ahmad Bilal.

**Writing – review & editing:** Norma Latif Fitriyani, Yeonghyeon Gu, Muhammad Syafrudin.

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
