## [Decision Letter · Decision Letter 0]

16 Sep 2024

PONE-D-24-30954Novel Deep Neural Network Architecture Fusion to Simultaneously Predict Short-Term and Long-Term Energy ConsumptionPLOS ONE

Dear Dr. Syafrudin,

Thank you for submitting your manuscript to PLOS ONE. After careful consideration, we feel that it has merit but does not fully meet PLOS ONE’s publication criteria as it currently stands. Therefore, we invite you to submit a revised version of the manuscript that addresses the points raised during the review process.

We look forward to receiving your revised manuscript.

Kind regards,

Jinran Wu, PhD

Academic Editor

PLOS ONE

Journal Requirements:

2. Please note that PLOS ONE has specific guidelines on code sharing for submissions in which author-generated code underpins the findings in the manuscript. In these cases, all author-generated code must be made available without restrictions upon publication of the work.

Please review our guidelines at https://journals.plos.org/plosone/s/materials-and-software-sharing#loc-sharing-code and ensure that your code is shared in a way that follows best practice and facilitates reproducibility and reuse.

“This work was supported by Institute of Information & communications Technology Planning & Evaluation (IITP) grant funded by the Korea government(MSIT) (No.1711160571, MLOps Platform for Machine learning pipeline automation.)”

5. Please note that funding information should not appear in the Acknowledgments section or other areas of your manuscript. We will only publish funding information present in the Funding Statement section of the online submission form. Please remove any funding-related text from the manuscript.

7.  We note you have included a table to which you do not refer in the text of your manuscript. Please ensure that you refer to Table 7 in your text; if accepted, production will need this reference to link the reader to the Table.

Additional Editor Comments (if provided):

Please carefully revise the manuscript with two referees' comments.

Reviewers' comments:

Reviewer's Responses to Questions

**Comments to the Author**

1. Is the manuscript technically sound, and do the data support the conclusions?

Reviewer #1: Yes

Reviewer #2: Yes

2. Has the statistical analysis been performed appropriately and rigorously? 

Reviewer #1: Yes

Reviewer #2: Yes

3. Have the authors made all data underlying the findings in their manuscript fully available?

Reviewer #1: Yes

Reviewer #2: Yes

4. Is the manuscript presented in an intelligible fashion and written in standard English?

Reviewer #1: Yes

Reviewer #2: Yes

5. Review Comments to the Author

Reviewer #1: The manuscript addresses an important issue in energy consumption prediction by proposing a novel hybrid model that combines Convolutional Neural Networks (CNN), Long Short-Term Memory (LSTM), and Bi-directional LSTM (Bi-LSTM) architectures to predict both short-term and long-term residential energy consumption. While the approach is interesting and potentially valuable, there are several areas where the manuscript could be improved.

1. The paper claims to present a novel hybrid model for predicting energy consumption. However, the innovation appears to be relatively incremental given the existing literature on CNN, LSTM, and Bi-LSTM architectures.

2. The rationale for combining short-term and long-term predictions in a single model is not sufficiently justified. Why is it necessary to predict both simultaneously? What specific advantages does this approach offer over using separate models for short-term and long-term predictions?

3. The manuscript does not provide adequate details on how the data for different prediction horizons were obtained. Are these data sets collected from the same source? How do the authors handle the differences in data frequency and resolution?

4. The manuscript lacks sufficient data visualization to support its claims. Data visualizations such as time series plots, correlation matrices, or feature importance rankings would provide valuable insights into the data and the model's performance.

5. While the manuscript reports performance metrics such as Mean Square Error (MSE) and Mean Absolute Error (MAE), it does not provide a comprehensive comparison with baseline models or other state-of-the-art methods.

6. The evaluation section would benefit from additional details about the experimental setup, including the size of the training and testing datasets, the hyperparameters used for the models, and the computational resources required. Multiple experiments are also necessary.

7. Ensure that all acronyms and technical terms are clearly defined when first introduced to readers.

Reviewer #2: The manuscript presents a hybrid deep learning model integrating Convolutional Neural Networks (CNN), Long Short-Term Memory (LSTM), and Bi-directional LSTM (Bi-LSTM) for forecasting energy consumption. I have several suggestions and concerns that need to be addressed before the manuscript can be considered for publication.

1. Comparison with Decomposition Algorithms: The authors mention the use of Empirical Mode Decomposition (EEMD) in existing literature for daily energy consumption prediction. It is suggested that the manuscript would benefit from the inclusion of comparative experiments that analyze the performance of the proposed model against models incorporating decomposition algorithms like EMD, EEMD and CEEMDAN.

2. Theoretical Justification: The manuscript claims that the proposed model can effectively handle both short-term and long-term energy consumption predictions. However, a theoretical explanation is lacking on how the model achieves this dual capability. The authors should elaborate on the theoretical aspects and mechanisms that allow the model to address the complexities of short-term and long-term predictions simultaneously.

3. Rationale for Model Parameter Selection: Please provide the basis for the hyperparameters chosen for the models.

4. Ablation Study: To better understand the contribution of each component of the proposed model, an ablation study could be beneficial. The authors are encouraged to compare the performance of the model with different configurations, such as CNN+BiLSTM, CNN+LSTM, and BiLSTM+LSTM.

5. Figure 3 Clarifications and Enhancements: The visualization in Figure 3 is not clear and uses a mix of bar charts and line graphs without a clear rationale. The authors should reconsider the design of Figure 3 to ensure that it effectively communicates the results. Additionally, the figure should be redrawn with higher resolution to improve clarity.

6. Consistency in Comparative Models: There is an inconsistency in the models compared across Table 3, Table 4, and Table 5. The authors should provide a rationale for these differences in the comparison models.

6. PLOS authors have the option to publish the peer review history of their article (what does this mean?). If published, this will include your full peer review and any attached files.

Reviewer #1: No

Reviewer #2: No

---

## [Author Response · Author response to Decision Letter 0]

28 Oct 2024

Dear Editors and Reviewers,

We would like to extend our heartfelt gratitude for your valuable feedback and insightful suggestions, which have greatly contributed to the improvement of our paper. Your comments have provided us with valuable guidance and direction, and we truly appreciate your positive recommendation.

Please see the attachment and thank you.

---

## [Decision Letter · Decision Letter 1]

11 Nov 2024

PONE-D-24-30954R1Novel Deep Neural Network Architecture Fusion to Simultaneously Predict Short-Term and Long-Term Energy ConsumptionPLOS ONE

Dear Dr. Syafrudin,

Thank you for submitting your manuscript to PLOS ONE. After careful consideration, we feel that it has merit but does not fully meet PLOS ONE’s publication criteria as it currently stands. Therefore, we invite you to submit a revised version of the manuscript that addresses the points raised during the review process.

We look forward to receiving your revised manuscript.

Kind regards,

Jinran Wu, PhD

Academic Editor

PLOS ONE

Reviewers' comments:

Reviewer's Responses to Questions

**Comments to the Author**

1. If the authors have adequately addressed your comments raised in a previous round of review and you feel that this manuscript is now acceptable for publication, you may indicate that here to bypass the “Comments to the Author” section, enter your conflict of interest statement in the “Confidential to Editor” section, and submit your "Accept" recommendation.

Reviewer #1: (No Response)

Reviewer #3: All comments have been addressed

2. Is the manuscript technically sound, and do the data support the conclusions?

Reviewer #1: Yes

Reviewer #3: Partly

3. Has the statistical analysis been performed appropriately and rigorously? 

Reviewer #1: Yes

Reviewer #3: Yes

4. Have the authors made all data underlying the findings in their manuscript fully available?

Reviewer #1: Yes

Reviewer #3: Yes

5. Is the manuscript presented in an intelligible fashion and written in standard English?

Reviewer #1: Yes

Reviewer #3: Yes

6. Review Comments to the Author

Reviewer #1: 1. The abstract is overly verbose, with much of it spent summarizing previous literature rather than succinctly presenting the motivation and unique contributions of the research. The novelty is not clearly convincing, making it difficult for readers to grasp the specific advancements introduced.

2. The paper lacks logical coherence; for example, the dataset is introduced within the methodology section, which is misplaced. Additionally, the research motivation is presented only after the literature review and formulation of the research question, creating a confusing sequence. Reorganizing the content would improve clarity.

3. The order of the three models (CNN, LSTM, and Bi-LSTM) in the hybrid model lacks justification. An explanation for this specific order, as well as a consideration of alternative arrangements, would strengthen the methodology.

4. The figures are low quality, with inconsistent font sizes, and they lack essential comparative elements, such as prediction comparison curves, which would be more informative than the current error convergence graphs. Adding these would improve the visual relevance and readability.

5. Lastly, the English language quality is poor, with significant grammatical and syntactical issues throughout. Professional editing or proofreading is strongly recommended to enhance the readability and professionalism of the manuscript.

Reviewer #3: Dear Editor,

Thank you for the opportunity to review this article. Overall, I believe the topic discussed in this paper is very meaningful, and the structure is fairly clear. The model proposed by the authors effectively addresses both long-term and short-term energy consumption issues, which holds significant value for time-series forecasting problems. Below are some of my personal comments for reference:

1.The evaluation results are confusing: It appears that different models were used when testing with data of varying levels, which is somewhat perplexing. Typically, it is expected to use the same set of models for comparison.

2.The model comparison is not comprehensive enough: The proposed model combines CNN, LSTM, and Bi-LSTM. As part of the ablation study, adding evaluations of Bi-LSTM would further validate the significance of each component in the proposed model architecture. Additionally, the attention mechanism is widely used in time series prediction tasks, and readers would be interested in a performance comparison between the proposed method and attention-based models.

3.Flaws in the design of the comparative experiment: the authors considered that their proposed model performed well in both long-term and short-term forecasting. However, the experimental design appeared to have only adjusted the time granularity without changing the sequence length, which could not intuitively demonstrate the model’s effectiveness on both long and short sequences.

4.The description of the ablation experiments is not sufficiently clear: the paper used data at different levels, but it was not explicitly stated in the ablation experiment whether the data used corresponds to a specific level or was an aggregated result across different levels. Additionally, including the baseline results in Table 9 would facilitate easier comparison.

5.Limited discussion on computational complexity: The authors didn’t discuss the computational complexity of their proposed model, which was crucial for evaluating its scalability and practical use in real-world applications.

6.The results in the charts are not sufficiently clear: the font in the images is too small. What does the black line in Figure 5 represent? Why are only some features labeled?

Recommendations:

1.In evaluations based on data of different levels, try to use only the same type of models. Include the performance of the baseline model Bi-LSTM as well as attention-based models.

2.Adding an evaluation of the model's predictive performance across different sequence lengths could better demonstrate the robustness of the proposed model.

3.Refine the description of the ablation experiments and highlight the conclusion that each component of the model’s three architectures (CNN, LSTM, and Bi-LSTM) is meaningful.

4.Add an evaluation of the model's operational efficiency or computational complexity.

5.Enlarge the font in the image results, and provide explanations for any special symbols.

6.Notably, the performance of the model proposed by the authors, especially in terms of MSE, was hundreds of times lower than that of other common time series prediction models. If there are no issues, such as recording errors, in the experimental results, this would represent a significant breakthrough. It is recommended that the authors submit their code to verify the authenticity of the experimental results.

In summary, the ideas in this paper are highly creative, and the structure is relatively clear. If the authors agree with and address the issues mentioned above, I recommend it for publication.

---

## [Author Response · Author response to Decision Letter 1]

26 Nov 2024

Response to Comments

Manuscript ID: PONE-D-24-30954R1

Title: Novel Deep Neural Network Architecture Fusion to Simultaneously Predict Short-Term and Long-Term Energy Consumption

Dear Editor,

Thank you very much for allowing us to revise the manuscript. We would like to thank the editor and all the reviewers for their valuable comments and suggestions. Based on the feedback, we have revised our manuscript. The detailed modifications to address reviewers’ comments are provided in the following. For clarity, we have marked our responses in blue. Whenever we update a paragraph in the manuscript, we mark it as a red color.

Reviewer #1

Point 1. The abstract is overly verbose, with much of it spent summarizing previous literature rather than succinctly presenting the motivation and unique contributions of the research. The novelty is not clearly convincing, making it difficult for readers to grasp the specific advancements introduced.

Response: The authors are highly grateful for your efforts and insightful comments. We apologize for the inconvenience and inappropriate language that raised ambiguity. The paper is extensively revised to remove ambiguity.

As per your valuable suggestion, we have revised the abstract in the updated version of the manuscript as:

“Energy is integral to the socio-economic development of every country. This development leads to a rapid increase in the demand for energy consumption. However, due to the constraints and costs associated with energy generation resources, it has become crucial for both energy generation companies and consumers to predict energy consumption well in advance. Forecasting energy needs through accurate predictions enables companies and customers to make informed decisions, enhancing the efficiency of both energy generation and consumption. In this context, energy generation companies and consumers seek a model capable of forecasting energy consumption both in the short term and the long term. Traditional models for energy prediction focus on either short-term or long-term accuracy, often failing to optimize both simultaneously. Therefore, this research proposes a novel hybrid model employing Convolutional Neural Network (CNN), Long Short-Term Memory (LSTM), and Bi-directional LSTM (Bi-LSTM) to simultaneously predict both short-term and long-term residential energy consumption with enhanced accuracy measures. The proposed model is capable of capturing complex temporal and spatial features to predict short-term and long-term energy consumption. CNNs discover patterns in data, LSTM identifies long-term dependencies and sequential patterns and Bi-LSTM identifies complex temporal relations within the data. Experimental evaluations expressed that the proposed model outperformed with a minimum Mean Square Error (MSE) of 0.00035 and Mean Absolute Error (MAE) of 0.0057. Additionally, the proposed hybrid model is compared with existing state-of-the-art models, demonstrating its superior performance in both short-term and long-term energy consumption predictions.”

Point 2. The paper lacks logical coherence; for example, the dataset is introduced within the methodology section, which is misplaced. Additionally, the research motivation is presented only after the literature review and formulation of the research question, creating a confusing sequence. Reorganizing the content would improve clarity.

Response: We again apologize for the inconvenience and inappropriate language that raised ambiguity. As per your valuable suggestion, we have reorganized the content in the updated version of the manuscript.

Point 3. The order of the three models (CNN, LSTM, and Bi-LSTM) in the hybrid model lacks justification. An explanation for this specific order, as well as a consideration of alternative arrangements, would strengthen the methodology.

Response: As per your valuable suggestion we have provided the justification regarding the order of the three models (CNN, LSTM, and Bi-LSTM) as:

For the first time, we have combined the three neural network layers (CNN, LSTM, and Bi-LSTM) for predicting energy consumption and achieved high performance compared to state-of-the-art studies.

In addition, the specific justification regarding the order of the three models is highlighted in the following section:

1. Novel Layers fusion of model’s architectures:

Recent research separately used CNN, LSTM, and Bi-LSTM models, where our proposed hybrid model integrates three models for energy consumption prediction. This unique hybridization allows proposed model to capture spatial and temporal features of input data which make it possible to handle energy consumption behavior from different perspectives that might be ignored when a single model is employed.

2. Enhanced Feature Learning:

CNN layer is deployed firstly in the architecture which allow hybrid model to extract local patterns and outliers in energy consumption data. This is usually most suitable for very short-term energy consumption prediction where the consumption pattern may quickly change. Recent studies focus on LSTM and Bi_LSTM for temporal prediction and neglecting the convolutional layers in feature extractions.

3. Bidirectional Processing:

Bi-LSTM is important to improve accuracy and it has ability to capture both future and past contexts in the energy consumption time series dataset. In recent studies very rarely Bi-LSTM is integrated with CNN which provides a novel angel to our study.

3. Tackling Present Constraints and Other Obstacles:

Proposed model overcome the limitations in existing literature such as inabilities to generalize across short-term and long-term energy consumption prediction simultaneously. Proposed model fills these gaps up contributes toward the existing body of knowledge in energy consumption prediction as a considerable addition.

In summary, according to the appraisal of the available contemporary resources on the subject, even though the proposed novel hybrid model features several components present in the literature, the combined use of CNN, LSTM, and Bi-LSTM for the task of energy consumption forecasting and its thorough testing on practical data is genuinely new and extends its usefulness within real-world situations. We feel these revolutions ought to be looked into more thoroughly and stress upon the relevance of our study towards the improvement of energy prediction methods and their relevance to the existing trends.

Point 4. The figures are low quality, with inconsistent font sizes, and they lack essential comparative elements, such as prediction comparison curves, which would be more informative than the current error convergence graphs. Adding these would improve the visual relevance and readability.

 Response: As per your valuable suggestion we have revised the figures in the updated version of manuscript. In addition, we have added the prediction comparison curves as:

Point 5. Lastly, the English language quality is poor, with significant grammatical and syntactical issues throughout. Professional editing or proofreading is strongly recommended to enhance the readability and professionalism of the manuscript.

Response: We again apologize for the inconvenience and inappropriate language that raised ambiguity. The paper is extensively revised to remove ambiguity.

Reviewer #3

Thank you for the opportunity to review this article. Overall, I believe the topic discussed in this paper is very meaningful, and the structure is fairly clear. The model proposed by the authors effectively addresses both long-term and short-term energy consumption issues, which holds significant value for time-series forecasting problems. Below are some of my personal comments for reference:

Point 1. The evaluation results are confusing: It appears that different models were used when testing with data of varying levels, which is somewhat perplexing. Typically, it is expected to use the same set of models for comparison.

Response: The authors are highly grateful for your efforts and insightful comments. We apologize for the inconvenience and inappropriate language that raised ambiguity. The paper is extensively revised to remove ambiguity.

We appreciate your observation regarding the evaluation process. To clarify, while it may appear that different models were used for testing at varying levels, we assure you that the same dataset, the IHEPC dataset, spanning a four-year period (2006–2010) and comprising 2,075,259 records with 12 attributes, was consistently used throughout our research.

Our approach involved evaluating the proposed model across three distinct temporal granularities hourly, daily, and monthly data. This was done to thoroughly assess the model's adaptability and performance at various levels of aggregation, which is crucial for understanding energy consumption patterns in diverse scenarios.

Point 2. The model comparison is not comprehensive enough: The proposed model combines CNN, LSTM, and Bi-LSTM. As part of the ablation study, adding evaluations of Bi-LSTM would further validate the significance of each component in the proposed model architecture. Additionally, the attention mechanism is widely used in time series prediction tasks, and readers would be interested in a performance comparison between the proposed method and attention-based models.

Response: As per your valuable suggestion we have revised the ablation study and added the attention-based model’s comparisons in the updated version of the manuscript as:

“Ablation Study

In our ablation study, we conducted a comparative analysis to investigate the performance of hybrid deep learning models. The results, as detailed in Table 9 for each model. The CNN-LSTM model yielded an MSE of 0.095, RMSE of 0.309, MAE of 0.238, and MAPE of 31.84\\%, indicating a reasonable predictive performance. However, the CNN+Bi-LSTM model, with slightly higher error metrics, did not significantly outperform the CNN-LSTM. Notably, the BiLSTM+LSTM configuration achieved superior accuracy, with minimal errors across all metrics. This significant reduction in errors suggests that BiLSTM+LSTM is highly effective for this task, capturing complex patterns more accurately compared to the other configurations.”

Attention-based model’s comparisons:

“Performance summarization with baseline models

Table 8 summarizes the average performance of the proposed model compared to similar techniques in the literature. The values are computed by averaging minute, hourly, daily and weekly results. Across these intervals, the proposed model consistently outperforms others based on four standard evaluation measures of MSE, RMSE, MAE and MAPE. This superiority extends to minute, hourly, daily and weekly data from the IHEPC dataset. This work can enhance intelligent power management systems, especially in predicting future electric power consumption. The average results are visualized in Figure 6.”

Point 3. Flaws in the design of the comparative experiment: the authors considered that their proposed model performed well in both long-term and short-term forecasting. However, the experimental design appeared to have only adjusted the time granularity without changing the sequence length, which could not intuitively demonstrate the model’s effectiveness on both long and short sequences.

Response: We acknowledge the concern regarding the design of the comparative experiment. While we adjusted the time granularity to evaluate the model's performance across different forecasting horizons, we recognize that varying sequence lengths could provide a more comprehensive assessment of its effectiveness. In response, we have revised the experimental setup to include evaluations using both short and long input sequences, ensuring a clearer demonstration of the model’s capability in handling diverse sequence lengths for both long-term and short-term forecasting. We appreciate your insightful comment and have incorporated this improvement in the revised manuscript.

Point 4. The description of the ablation experiments is not sufficiently clear: the paper used data at different levels, but it was not explicitly stated in the ablation experiment whether the data used corresponds to a specific level or was an aggregated result across different levels. Additionally, including the baseline results in Table 9 would facilitate easier comparison.

Response: As per your valuable suggestion we have revised the description of the ablation experiments in the updated version of the manuscript as:

“Ablation Study 

In our ablation study, we conducted a comparative analysis to investigate the performance of hybrid deep learning models using a complete dataset. The results, as detailed in Table 9 for each model. The CNN-LSTM model yielded an MSE of 0.095, RMSE of 0.309, MAE of 0.238, and MAPE of 31.84\\%, indicating a reasonable predictive performance. However, the CNN+Bi-LSTM model, with slightly higher error metrics, did not significantly outperform the CNN-LSTM. Notably, the BiLSTM+LSTM configuration achieved superior accuracy, with minimal errors across all metrics. This significant reduction in errors suggests that BiLSTM+LSTM is highly effective for this task, capturing complex patterns more accurately compared to the other configurations.”

Point 5. Limited discussion on computational complexity: The authors didn’t discuss the computational complexity of their proposed model, which was crucial for evaluating its scalability and practical use in real-world applications.

Response: As per your valuable suggestion we have added the computational complexity in the updated version of the manuscript as:

“Computational complexity

The computational complexity of the proposed model in Table 9 demonstrates its efficiency across varying temporal granularities. Using the IHEPC dataset, the model required 486 seconds for daily data, 119 seconds for weekly data, and just 32 seconds for hourly data. This performance highlights the model's scalability and suitability for both coarse-grained and fine-grained energy consumption analysis, ensuring practical usability across diverse applications.”

Point 6. The results in the charts are not sufficiently clear: the font in the images is too small. What does the black line in Figure 5 represent? Why are only some features labeled?

Response: The black line in Figure 5 represents the error values. Since this graph is based on feature importance, we have selected the most important features and visualized their importance values.

Recommendations:

Point 1. In evaluations based on data of different levels, try to use only the same type of models. Include the performance of the baseline model Bi-LSTM as well as attention-based models.

Response: As per your valuable suggestion we have included the performance of the baseline model Bi-LSTM as well as attention-based models in the updated version of the manuscript as:

“Performance summarization with baseline models

Table 8 summarizes the average performance of the proposed model compared to similar techniques in the literature. The values are computed by averaging minute, hourly, daily and weekly results. Across these intervals, the proposed model consistently outperforms others based on four standard evaluation measures of MSE, RMSE, MAE and MAPE. This superiority extends to minute, hourly, daily and weekly data from the IHEPC dataset. This work can enhance intelligent power management systems, especially in predicting future electric power consumption. The average results are visualized in Figure 6.”

Point 2. Adding an evaluation of the model's predictive performance across different sequence lengths could better demonstrate the robustness of the proposed model.

Response: As per your valuable suggestion we have revised the evaluation of the model's predictive performance across different sequences in the updated version of the manuscript.

Point 3. Refine the description of the ablation experiments and highlight the conclusion that each component of the model’s three architectures (CNN, LSTM, and Bi-LSTM) is meaningful.

Response: As per your valuable suggestion we have revised the ablation study and highlighted the conclusion in the updated version of the manuscript as:

“Ablat

---

## [Editor Report · Decision Letter 2]

29 Nov 2024

Novel Deep Neural Network Architecture Fusion to Simultaneously Predict Short-Term and Long-Term Energy Consumption

PONE-D-24-30954R2

Dear Dr. Muhammad Syafrudin,

We’re pleased to inform you that your manuscript has been judged scientifically suitable for publication and will be formally accepted for publication once it meets all outstanding technical requirements.

Kind regards,

Jinran Wu, PhD

Academic Editor

PLOS ONE

---

## [Editor Report · Acceptance letter]

14 Dec 2024

PONE-D-24-30954R2 

PLOS ONE

Dear Dr. Syafrudin, 

I'm pleased to inform you that your manuscript has been deemed suitable for publication in PLOS ONE. Congratulations! Your manuscript is now being handed over to our production team.

Kind regards, 

on behalf of

Dr. Jinran Wu 

Academic Editor

PLOS ONE